# Confidence and Error Analyses of the Radiosonde and Ka-Wavelength Cloud Radar for Detecting the Cloud Vertical Structure

**Yun Yuan, Huige Di \*, Yuanyuan Liu, Danmin Cheng, Ning Chen, Qing Yan and Dengxin Hua**

School of Mechanical and Precision Instrument Engineering, Xi'an University of Technology, Xi'an 710048, China
* Correspondence: dihuige@xaut.edu.cn

**Abstract:** A macro-vertical structure is closely related to weather evolution and the energy budget balance of the atmospheric system of the Earth. In this study, radiosonde data were used to identify a cloud vertical structure (CVS) using the adjusted relative humidity threshold method. To evaluate the reliability and stability of this method, the results obtained based on the spatiotemporal matching criteria established in this study were compared with Ka-band millimetre-wave cloud radar (MMCR) observation data. This comparison showed that both devices exhibit high consistency in low-level cloud detection. With the increase in the cloud height, the frequency of the cloud appearance detection by the radiosonde became higher than that by the MMCR. In spring, the results of the CVS detection by the two devices were in good agreement. Specifically, the determination coefficients of the modified degrees of freedom (adjusted R-square) of the cloud base height (CBH) and cloud top height (CTH) detected by the two devices were 0.934 and 0.879, respectively. The horizontal drift of the radiosonde was the smallest in summer, and the adj. R-square values of the CBH and CTH were 0.814 and 0.852, respectively. The CVS observation results by the radiosonde and the MMCR were significantly different in autumn (the adj. R-Square values of the CBH and CTH were 0.715 and 0.629, respectively). In winter, the adj. R-Square values of the CBH and CTH observed by the radiosonde and the MMCR were 0.958 and 0.710, respectively. The statistics and analysis of the results of the distribution characteristics of the CVSs using radiosonde data from 2019 to 2021 from Xi'an showed that the average CTH and CBH were at 7–10 km and 3–5 km, respectively. The frequencies of the cloud absence, rainfall, and two- and three-layer clouds were the highest in the winter (34.36%), autumn (12.99%), and summer, respectively.

**Keywords:** radiosonde; Ka-band millimetre-wave cloud radar; cloud vertical structure; relative humidity (RH) threshold method

## 1. Introduction

Clouds are related to the density of atmospheric dynamic processes, thermal processes, the water vapor cycle, and energy budgets [1,2]. Cloud vertical structures (CVSs, including the cloud base height, cloud top height, cloud layer number, and cloud types) and the distribution of multi-layer clouds in the atmosphere affect atmospheric dynamics, thermo-dynamics, and the hydrological cycle. They also impact large-scale atmospheric circulation via radiative heating/cooling and latent heat release [3,4]. Cloud information changes rapidly across space and time, which makes their observation challenging. This, in turn, leads to considerable uncertainty in cloud-related climate forecasts [5]. To understand cloud physical processes and improve the prediction ability of large-scale climate models (including global circulation models), it is necessary to observe the CVS in a highly accurate manner.

Types of ground-based active remote sensing detection equipment, such as lidar, Ka-band millimetre-wave cloud radar (MMCRs), and cloud altimeters, are powerful tools for automatic cloud observation [6–9]. Lidars and cloud altimeters have been extensively used to determine the cloud base height (CBH); however, owing to the attenuation of beam

energy in a cloud [10,11], accurately detecting the cloud top height (CTH) is impossible. In contrast, an MMCR can continuously observe CVSs with high accuracy. However, it is subject to serious attenuation when encountering precipitation clouds [12]. Radiosondes can directly penetrate clouds and provide wind direction, wind speed, temperature, pressure, and humidity data from ground level up to an altitude of 30 km [13–15]. Radiosondes have many emission points worldwide and can realise the large-scale observation of clouds via networking. At present, the methods using radiosondes to observe CVSs include the temperature dew point difference [16], relative humidity (RH) threshold [17], and second derivative methods [18]. Wang and Rossow [12] found that the RH change in a radiosonde is closely related to cloud information. Based on the frequency statistics of the RH values within the CBH range observed from the ground by a radiosonde over one year, they proposed 84–87% as the threshold value. Moreover, they detected a CVS under the condition of a negative positive change in the RH generated by CTH and the CBH, which is called the RH threshold method (WR95 method). However, the WR95 method tended to misclassify moist cloudless atmospheric layers as clouds, and both radiosonde techniques reported higher cloud tops than those observed using the cloud radar. Zhang et al. [19] used radiosonde data (Vaisala radiosonde-RS92) obtained by the Atmospheric Radiation Measurement Mobile Facility at Shouxian, Anhui Province, China (116°27′–117°04′E, 31°54′–32°40′N, subtropical northern monsoon humid climate). They combined these data with the observation results of an MMCR and a cloud altimeter to improve the WR95 method. Although the influence of the horizontal drift of the radiosonde was considered, the reliability of the radiosonde detection under different weather conditions and different cloud types was not analysed. Therefore, when analysing the accuracy and reliability of the radiosonde in detecting the CVS, it is necessary to comprehensively consider the horizontal drift caused by the wind speed and wind direction of the radiosonde and the changes in different weather and different cloud types, so as to comprehensively evaluate the performance of the radiosonde in detecting the CVS.

Owing to global climate change, Xi'an, in China (a large inland city, 107°40′–109°49′E, 33°42′–34°45′N, warm temperate climate), frequently experiences high-temperature weather [20–22], which poses a threat to human security and economic growth. Although studies have examined the use of different detection equipment to evaluate the changes in the CBHs and the CTHs of clouds over Xi'an, analyses of the distribution and change characteristics of CVSs are still lacking, mainly because of the absence of long-term observation data and reliable CVS recognition algorithms. In this study, CVSs over Xi'an were studied using radiosonde and MMCR data from the Jinghe Meteorological Station. The effects of the cloud type, cloud height, drift, and other factors on the CVS observations by the radiosonde in different weather conditions were comprehensively analysed. By comparing and analysing the radiosonde results with those of the MMCR (from December 2020 to November 2021), the reliability of using a radiosonde to detect a CVS was evaluated. The change characteristics of the CVSs over Xi'an were analysed based on the radiosonde data from 2019 to 2021. Thus, this study provides relevant information about the cloud measurement ability of radiosondes and MMCRs, joint observations, and research on the characteristics of climate change in Xi'an.

## 2. Method

### 2.1. Cloud Detection by Radiosonde

The WR95 method uses rawinsonde data to estimate the cloud vertical structure, including the cloud top and cloud base heights, cloud layer thickness, and the characteristics of multi-layered clouds. At the cloud layer base, a minimum relative humidity of at least 84% has been observed, with relative humidity increases exceeding 3% at the cloud layer top and base, where the relative humidity varies with respect to the liquid water at temperatures greater than or equal to 0 °C and with respect to ice at temperatures less than 0 °C [1]. In 2010, Zhang et al. [19] improved the WR95 method (i.e., to ZHA10 method) and established linear RH thresholds (min-RH, Max-RH, and inter-RH) by varying the height. They

analysed the CVS detection by the Vaisala radiosonde (model is RS92) in Shouxian, China, and established the 'inter-RH' threshold as the condition for merging two adjacent humidity layers. The performance parameters of the CTS11 radiosonde at the Jinghe Meteorological Station (Station No. 57131, longitude: 108°58′E, latitude: 34°26′N, altitude: 411 m) are listed in Table 1. Considering the difference between the CTS11 and RS92 radiosondes, in this study we adjusted the RH threshold in ZHA10 to detect a CVS using the radiosonde in Xi'an. The adjusted cloud detection algorithm is presented in Table 2.

**Table 1.** Performance parameters of the GTS11 radiosonde.

| Measuring Performance | Temperature (°C) | Pressure (hPa) | Relative Humidity (%) |
|---|---|---|---|
| Measurement span | −90–50 | 1060–5 | 0–100 |
| Effective measurement span | −80–50 | 1050–10 | 10–90 |
| allowance error | $\Delta(T) \leq 0.3$ | Pressure $\geq$ 500 $\Delta(P) \leq 2$<br>Pressure < 500 $\Delta(P) \leq 1$ | $\Delta(RH) \leq 5$ |

**Table 2.** Height-resolving RH thresholds.

| Height Range (km) | Relative Humidity Threshold (%) | | |
|---|---|---|---|
| | Min RH | Max RH | Inter-RH |
| 0–2 | 92–90 | 95–93 | 84–82 |
| 2–6 | 90–88 | 93–90 | 82–78 |
| 6–12 | 86.2–72.5 | 87.5–77.5 | 75.5–67.5 |

Note: the cloud height over Xi'an does not exceed the stratospheric bottom height.

Before conducting the tests, the RH was first transformed with respect to the ice instead of liquid water for all levels with temperatures below 0 °C [23]. Subsequently, we performed examinations to identify the cloud layers in eight steps: (1) From the bottom to the top of an RH profile, if the RH was > min RH, the corresponding cloud height was regarded as the base height of the wet layer. (2) If the RH corresponding to the height above a wet layer was continuously greater than the min RH, that part was regarded as the same layer. (3) If the RH dropped below the min RH or exceeded it in the top layer of a profile, this layer was regarded as the top height of the wet layer. (4) A wet layer with a base height of less than 300 m and a thickness of less than 400 m was discarded. (5) If the distance between two wet layers was less than 300 m, or the minimum RH over this distance was more than the inter-RH in these two layers, the two layers were merged into one layer. (6) If the maximum RH in a wet layer was greater than the corresponding max RH at the base of the wet layer, the wet layer was identified as a cloud layer.

*2.2. Cloud Detection by MMCR*

The MMCR used in this study is located at the Jinghe Meteorological Station, which is equipped with various meteorological observation instruments (e.g., rain gauges, microwave radiometers, and lidars). The MMCR can output primary properties, such as the reflectivity factor, radial velocity, and velocity spectrum width. It has a vertical resolution of 30 m, temporal resolution of 5s for a single profile, maximum detection height of 15 km, and detection ability ranging from −40 dBZ to +40 dBZ. We used the reflectivity factor recorded by the MMCR to obtain the CTH, CBH, cloud thickness, and number of cloud layers. Ideally, the echo information of an MMCR only reflects the changes in the cloud information. However, owing to the stability of the working state of a radar, a signal-processing algorithm is required. Moreover, non-cloud targets produce interference and clutter, generating non-cloud echo interference signals [24,25]. Owing to the high space–time resolution of an MMCR, a single-layer cloud with a loose structure is misidentified as a multi-layer cloud. The interference of the MMCR implementation introduces considerable uncertainty to the determination of a CVS. Therefore, the quality control of cloud information must be performed as the last step, so as to improve the accuracy of the

cloud boundary determination. This includes two main aspects. The first is the elimination of interference signals. Based on the one-year observation data of the MMCR, this study determined the data quality control threshold by analysing the characteristic changes in the floating echo in the boundary layer under non-cloud conditions. Subjectively, when the echo intensity was Z < −20 dBZ, the absolute value of the radial velocity of <0.2 m·s$^{-1}$ and spectral width of >0.3 m·s$^{-1}$ were used as the threshold to eliminate the non-cloud information and interference signals. The second aspect was the merging and removal of loosely structured clouds and the calculation of the thickness of each cloud layer and the distance intervals between adjacent clouds. For clouds with thicknesses of less than 210 m, it was determined whether the distance between this cloud layer and its upper and lower cloud layers exceeded 720 m [26]. Clouds that satisfied these conditions were not considered in this study. Alternatively, a cloud layer meeting these conditions was combined with the nearest adjacent layer.

### 2.3. Spatiotemporal Matching Criteria

The spatial distribution of clouds is non-uniform, and the vertical velocityn fluctuates at the base and top of a cloud. Even for the same cloud, the CVS detection differs significantly at different locations. To compare the CVS detection by the radiosonde, whose rising track is affected by the wind speed and direction, with that by the fixed-point vertical observations using the MMCR, the data were configured in time and space. This configuration was estimated according to the rising time of the radiosonde so as to reduce the impact of the track change in the rising process of the radiosonde on the CVS detection. The average CBH and CTH during the rising period of the radiosonde were calculated as a result of the MMCR detecting the cloud base and top. The spatiotemporal matching criteria were as follows: a height range of 0–2500 m and the average cloud boundary height within 2 min before and after the launch time of the radiosonde (07:13–7:17 China Standard Time (CST) or 19:13–19:17 CST), which were used as the CBH and CTH detected by the MMCR. Similarly, within 2500–6000 m, the average cloud boundary height within 7 min before and after the launch time (07:08–7:22 CST or 19:08–19:22 CST) was used as the detection result of the MMCR. Above 6000 m, the average CBH and CTH obtained from the reflectivity factor at the cloud boundary during 7:00–7:59 CST or 19:00–19:59 CST were used as the MCCR detection results.

## 3. Typical Case Analysis

From the samples observed by the radiosonde and the MMCR in the same period, typical cases were selected according to the cloud type (low, middle, and high clouds, with 500 m ≤ CBH < 2000 m, 2000 m ≤ CBH < 6000 m, and CBH ≥ 6000 m, respectively) [27], the number of layers (two- and three-layer clouds), and precipitation. The rationality and reliability of the adjusted radiosonde RH thresholds were verified by analysing the results of the CVS detection by the two devices.

### 3.1. Case 1: CVSs of Low, Middle, and High Clouds

Figure 1 shows the RH profiles obtained by the radiosonde and the time-series intensity information (THI) of the reflectivity factor recorded by the MMCR on 13 October 2021. Using the RH threshold method to identify clouds based on the RH information obtained by the radiosonde, the estimated CBH, CTH, and cloud layer thickness were approximately 1058 m, 1923 m, and 865 m, respectively. The reflectivity factor observed by the MMCR showed that the cloud layer during the observation period was a low-level stratiform cloud with a smooth boundary. The height of the cloud layer was 1030–1830 m, and the cloud thickness was approximately 800 m. Comparing the two detection results, the low-level CVSs observed by the radiosonde and the MMCR were similar.

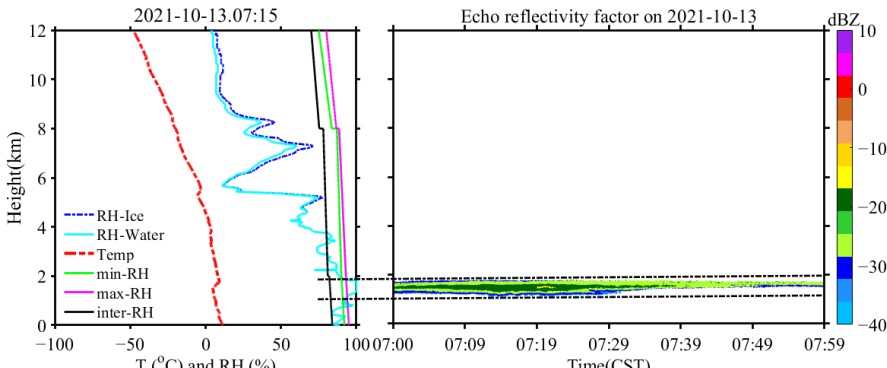

**Figure 1.** Low cloud detected by the radiosonde and MMCR on 13 October 2021 (radiosonde temperature and RH profile (**left**), reflectance factor THI from the MMCR (**right**)).

On 28 April 2021, the radiosonde and MMCR detected a middle cloud simultaneously, and the results are shown in Figure 2. The RH profiles obtained by the radiosonde show that the cloud was located between 4124 m and 6060 m, and its thickness was approximately 1936 m. The recorded reflectivity factor shows that the cloud boundary underwent significant changes with the increasing observation time. These changes were also accompanied by a continuous change in the cloud thickness from 400 m to 2600 m. Based on the spatiotemporal matching criteria, the cloud height observed by the MMCR was 3800–5400 m and the thickness was 1600 m. The CTH determined by the radiosonde based on the RH threshold method was higher than that determined by the MMCR detection.

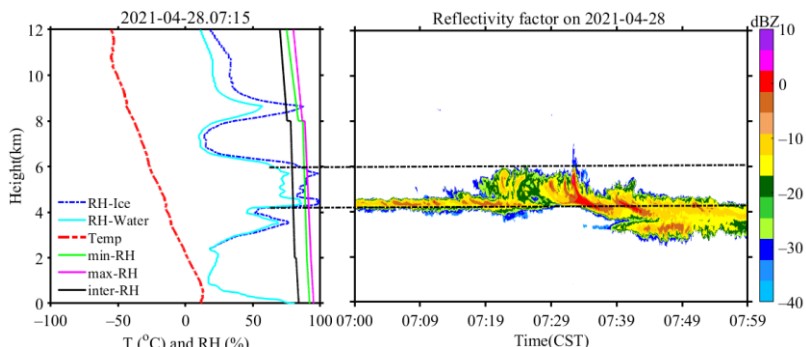

**Figure 2.** Middle cloud detected by the radiosonde and MMCR on 28 April 2021 (radiosonde temperature and RH profile (**left**), reflectivity factor THI from the MMCR (**right**)).

Figures 3 and 4 show two different types of high clouds observed by the radiosonde and the MMCR. The reflectance factor observed by the MMCR shows that, from 07:00 CST to 07:59 CST, on 24 January 2021, the clouds were stratiform clouds with flat cloud boundaries and a thickness of approximately 3900 m, within a range of 7200–11,070 m. During the same period, the cloud detected by the radiosonde was located between 7596 m and 11,290 m. For this high-level cloud, the cloud information observed by the radiosonde was significantly greater than that observed by the MMCR. The difference between the CBHs observed by the radiosonde and the MMCR may be caused by the response of the radiosonde after entering the cloud layer, resulting in a 426 m deviation in the CBH. The difference in the CTH may be due to the small particle size of the cloud top and the failure of the MMCR detection [28], resulting in the MMCR underestimating the CTH by 220 m. The temperature above the cloud top may have also been extremely low ($< -50\ ^\circ$C), and the response lag of the humidity sensor on the radiosonde could have led to the CTH overestimation by 220 m.

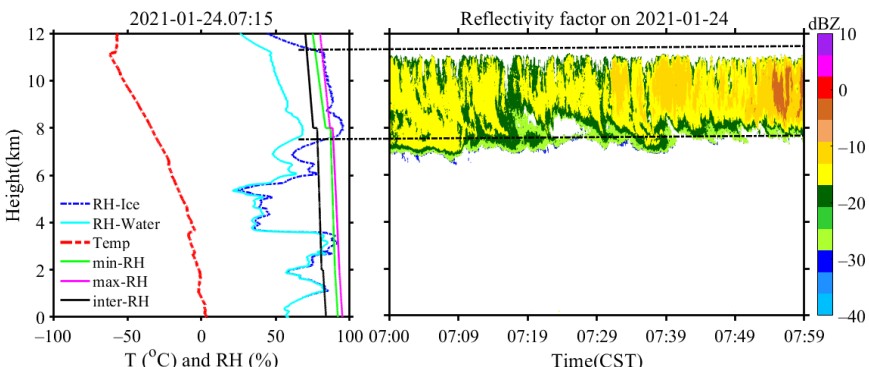

**Figure 3.** High cloud detected by the radiosonde and MMCR on 24 January 2021 (radiosonde temperature and RH profile (**left**), reflectivity factor THI from the MMCR (**right**)).

Based on the cloud reflectance factor observed by the MMCR, as shown in Figure 4, initially, the CBH was 8460 m and the CTH was 10,350 m. With the increasing observation time, a remarkable phenomenon involving the airflow convection in the environment was observed at the cloud top, which caused the cloud to gradually become thicker, and the highest part of the cloud top was 11,520 m. The cloud subsequently dissipated rapidly, and its thickness was only 600 m at the end of the observation period. However, the RH change obtained by the radiosonde cannot show any information about the cloud changes in this period. It is speculated that the radiosonde failed to observe the cloud information mainly because of its horizontal drift in the rising process (the cloud dissipated rapidly from 7:39 CST to 7:59 CST). This would have allowed the radiosonde to pass through the gap in the dissipated cloud and, therefore, it would not have detected any cloud information.

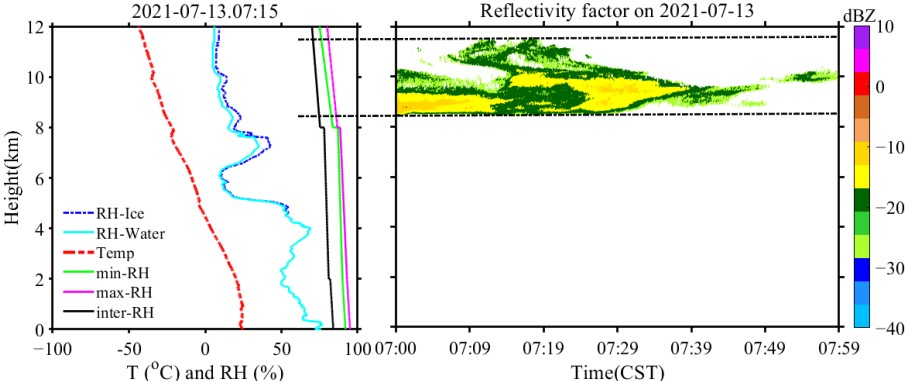

**Figure 4.** High cloud detected by the radiosonde and MMCR on 13 July 2021 (radiosonde temperature and RH profile (**left**), reflectivity factor THI from the MMCR (**right**)).

### 3.2. Case 2: Two-Layer Cloud

Figure 5 shows the results for a two-layer cloud structure observed by the MMCR and the radiosonde. The CBH and CTH of the lower layer (located between 1020 m and 3060 m) obtained by the radiosonde and the MMCR show good agreement. For the high-level cloud, based on the detection results of the radiosonde and the MMCR, the CBHs were similar (the CBH was approximately 5060 m), and the CTHs were 9735 m and 8867 m, respectively. Specifically, the deviation between the CTHs detected by the two devices was 868 m. The MMCR showed that the echo reflectivity factor around the cloud top was approximately −27 dBZ. We speculate that small ice particles may have been located on the cloud top, which cannot be detected by the MMCR, resulting in the underestimation of the CTH. Simultaneously, the humidity sensor of the radiosonde has a certain response delay above 7440 m (the height corresponding temperature is <−50 °C). Thus, it recorded an RH of approximately 88–80% within the range of 7440–9735 m, which was incorrectly recognised as cloud information.

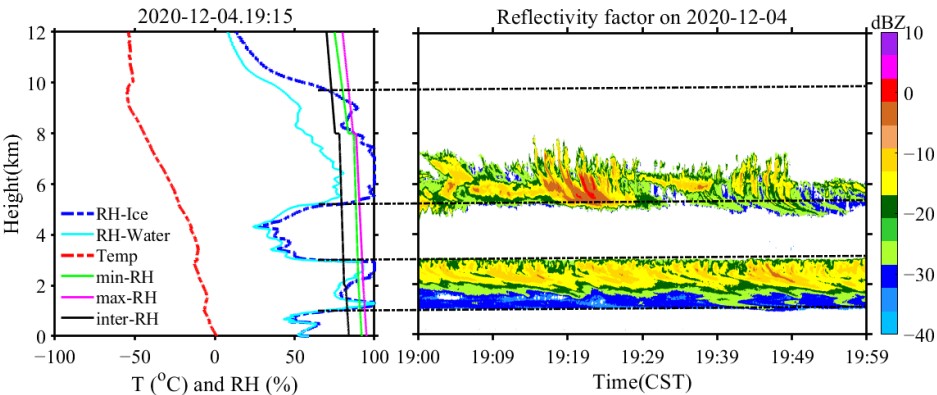

**Figure 5.** Two-layer cloud detected by the radiosonde and MMCR on 4 December 2021 (radiosonde temperature and RH profile (**left**), reflectivity factor THI from the MMCR (**right**).

### 3.3. Case 3: Three-Layer Cloud

Multi-layer clouds have a significant impact on the radiant heating or cooling of the atmosphere and the surface of the Earth [29]. Furthermore, they are useful cases for verifying the use of the RH threshold method to identify a CVS from RH profiles recorded by a radiosonde. As shown in Figure 6, from 07:00 CST to 07:59 CST, both the radiosonde and MMCR detected a three-layer cloud structure. The cloud structure information detected by the radiosonde was as follows: low-, middle-, and high-level clouds were located in the ranges of 3258–4900 m, 6363–6990 m, and 8624–10,460 m, respectively. The MMCR observation results show that the average heights of the low-, middle-, and high-level clouds were 3000–4800 m, 6210–6850 m, and 8494–9600 m, respectively. From the observation results of the radiosonde and the MMCR, the two devices showed strong consistency in detecting the CVSs of the low- and middle-level clouds, whereas the observation difference in the CTHs of the high-level cloud was still similar to that discussed in Section 3.2. However, in this case, the reflectivity factors of the small particles on the top of the high-level cirrus were smaller than −25 dBZ. The CTH overestimation due to the delayed response of the humidity sensor of the radiosonde was preliminarily eliminated (−50 °C corresponds to a height of 11,500 m). Consequently, the difference in the CTHs of the high-level cloud was underestimated by the MMCR, owing to its limited detection sensitivity.

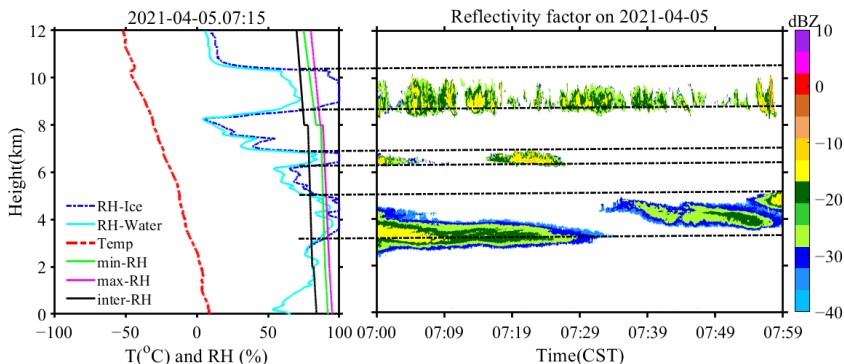

**Figure 6.** Three-layer cloud detected by the radiosonde and MMCR on 5 April 2021 (radiosonde temperature and RH profile (**left**), reflectivity factor THI from the MMCR (**right**)).

### 3.4. Case 4: Precipitation Cloud

On 19 April 2021, the weather changed from overcast sky to light rain, and precipitation occurred during the observation period from 19:00 CST to 19:59 CST. Figure 7 shows that the reflectivity factor observed by the MMCR reached 10 dBZ, and the echo signal touched the ground. These phenomena reflect the occurrence of precipitation. Based on the reflectivity factor, the CBH and the CTH were 270 m and 6990 m, respectively. The RH

change obtained by the radiosonde showed a large amount of water vapour from 2000 m to 12,000 m, and the water vapour layer was extremely thick in the same period. Based on the cloud information obtained using the RH threshold, the cloud was located between 2188 m and 11,000 m. Precipitation clouds have a significant attenuative effect on MMCRs [30,31]. Therefore, the MMCR in this study could not obtain the actual CTH. However, the presence of a large amount of water vapour at high altitudes due to precipitation cannot be ignored. Over an area of 6999–11,000 m, the RH was greater than the RH threshold, resulting in its incorrect identification as cloud information. Compared with the MMCR, the radiosonde can provide an effective CBH in this case.

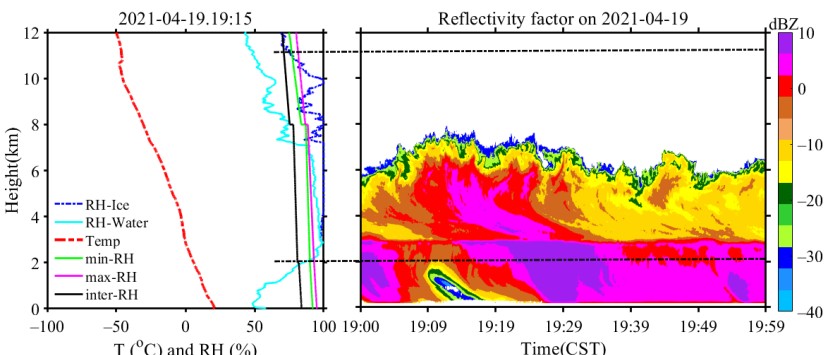

**Figure 7.** Precipitation cloud detected by the radiosonde and MMCR on 19 April 2021 (radiosonde temperature and RH profile (**left**), reflectivity factor THI from the MMCR (**right**)).

### 3.5. Case 5: Non-Precipitation Cloud

A light rainy weather changed to overcast conditions on 20 April 2021. As shown in Figure 8, the reflectivity factor obtained by the MMCR reached approximately ground level, most of the echo signal strengths were in the range of −30−−15 dBZ, and the average CTH was approximately 5670 m (the cloud top boundary was relatively flat). The CBH and CTH detected by the radiosonde were 500 m and 10,300 m, respectively. Combined with the reflectivity factor from 18:00 to 18:59 CST (checking the rainfall time recorded by a microwave radiometer at the same site, we observed that rainfall occurred from 18:00 to 18:39 CST), rainfall occurred before the launch of the radiosonde (19:15 CST). It can be inferred that the CTH detected by the MMCR was lower than that by the radiosonde mainly because of the large amount of water vapour within the height range of 5670–11,300 m. This led to a continuous response of the humidity sensor and, therefore, to the overestimation of the CTH by 1000 m.

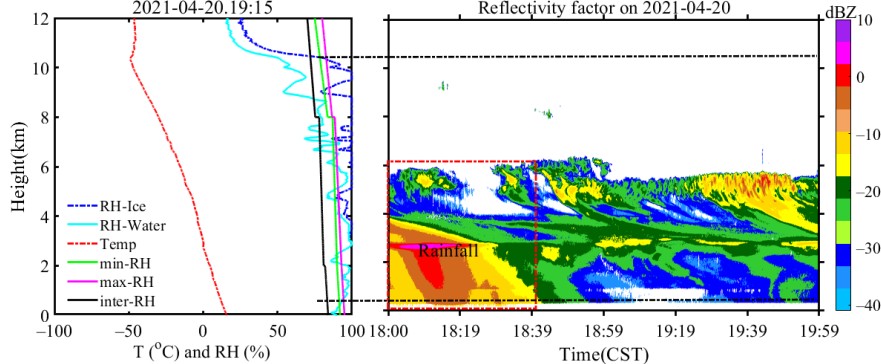

**Figure 8.** Non-precipitation cloud detected by the radiosonde and MMCR on 20 April 2021 (radiosonde temperature and RH profile (**left**), reflectivity factor THI from the MMCR (**right**)). (Red box indicates rainfall period detected by microwave radiometer.)

## 4. Analysis and Discussion of CVS Results

### 4.1. Observation Sample Statistics

We used the observation data of the radiosonde and the MMCR recorded from December 2020 to November 2021 (a total of 365 days) to calculate the CVS information, and we compared and analysed the detection results of the two devices. The radiosonde was launched twice a day at 07:15 a.m. and 19:15 p.m., with a sampling period of 1 s and a rising speed of 6–7 m·s$^{-1}$. Before comparing the CVSs detected by the radiosonde and the MMCR, we selected the observation data taken when the rising height of the radiosonde was $\geq$ 12 km and recorded the wind speed and direction. The MMCR data from 7:00–07:59 CST and 19:00–19:59 CST were selected to obtain the CBH and the CTH (the radiosonde was able to rise above 20 km in this period). Ideally, there should have been 730 groups of data for 365 days. However, owing to certain conditions of the equipment, the invalid data of the MMCR and the radiosonde produced 194 and 22 groups, respectively. Finally, 514 groups of effective data were obtained for the same period.

Based on the spatiotemporal matching criteria in Section 2.3, among the 514 observation samples, 395 were 'completely consistent' (76.84%), 38 samples were 'approximately consistent' (7.39%), and 81 samples were 'completely inconsistent' (15.7%). Excluding the samples including cloud-free and precipitation clouds, the completely consistent sample size was 222, with 38 and 81 samples partially and completely inconsistent, respectively. The specific percentage of each condition and the sample distribution in each quarter are shown in Figure 9.

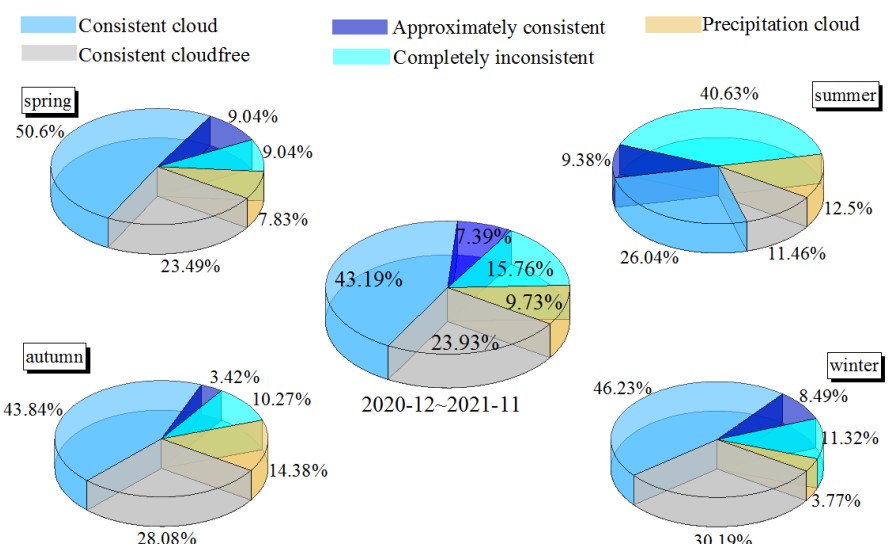

**Figure 9.** Percentages of the sample sizes in different cases: completely consistent (radiosonde and MMCR detect cloud-free and cloud conditions simultaneously), approximately consistent (numbers of cloud layers detected by radiosonde and MMCR are different), and completely inconsistent (radiosonde detects cloud and MMCR detects cloud-free condition, or MMCR detects cloud and radiosonde identifies a cloud-free scenario).The **upper left**, the **upper right**, the **lower left** and the **lower right** corner are spring, summer, autumn and winter respectively, and the **middle** of the figure is the annual record from December 2020 to November 2021.

### 4.2. Distribution Characteristics of the CVS

#### 4.2.1. Cloud Layer Distribution

Using the data recorded by both the radiosonde and MMCR when observing the clouds (260 samples), the CVSs detected by each were analysed. Figure 10a shows the average CBH, CTH, and cloud geometric thickness values of one-, two-, and three-layer clouds at the vertical height determined from the RH information recorded by the radiosonde, using the RH threshold method. Figure 10b shows the distribution of the CVSs observed by the MMCR during the same period. In the vertical direction, from a macro perspective, the

average heights of the one-layer clouds detected by the radiosonde and the MMCR lay between those of the two- and three-layer clouds, whereas the geometric thicknesses of the one-layer clouds were relatively thicker than those of the latter. In a three-layer cloud structure, the geometric thickness of the top-layer cloud exceeds those of the middle- and low-layer clouds. This may be attributed to a top layer causing the cooling intensity of the long-wave radiation at the top of a lower-layer cloud to decrease, causing the geometric scale of the top-layer cloud to be larger than that of the low-layer cloud.

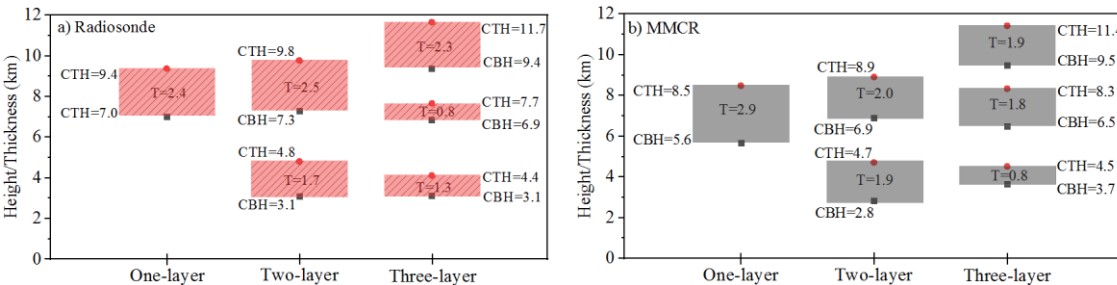

**Figure 10.** Vertical distributions of the average CBH, CTH, and cloud thickness. (**a**) CVSs observed by the radiosonde and (**b**) CVSs observed by the MMCR. T represents the geometric average thickness of the cloud.

The frequency distribution obtained by the observations of low-, middle-, and high-level clouds by the radiosonde and the MMCR is shown in Figure 11. The frequencies of low-, middle-, and high-level clouds detected by the radiosonde and MMCR were 11% and 13%, 37% and 41%, and 47% and 52%, respectively. These results show that both the radiosonde and the MMCR detected the lowest frequency of low-level clouds and the highest frequency of high-level clouds. With increasing cloud height, the difference between the frequencies of the cloud occurrence detected by the radiosonde and the MMCR gradually increased (the frequency differences for the low-, middle-, and high-level clouds were 2%, 4%, and 5%, respectively). This may be attributed to the change in the rising trajectory of the radiosonde, causing the detection of a cloud to differ from the detection by the MMCR. Alternatively, because the cloud particle size at the cloud top is far too small to be detected by the MMCR, the frequency of the cloud occurrence was underestimated. Therefore, further study of the abilities of the radiosonde and the MMCR to detect CVSs is necessary.

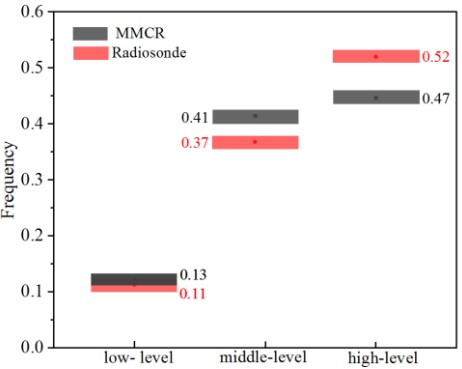

**Figure 11.** Frequencies of low-, middle- and high-level clouds observed by the radiosonde and MMCR.

### 4.2.2. Seasonal Distribution of Cloud Base and Top Heights

From December 2020 to November 2021, 260 cloud data samples were observed by the MMCR and the radiosonde. The four seasons were defined as follows: spring, from March to May (MAM), summer, from June to August (JJA), autumn, from September to November (SON), and winter, from December to February (DJF). Figure 12 shows the correlation between the CBHs and CTHs observed by the radiosonde and the MMCR in the four seasons. By analysing the former correlation, it can be inferred that the radiosonde

and MMCR results of the CBH in spring and winter are relatively consistent, with the adj. R-square values of 0.934 and 0.958, respectively. In summer, the adj. R-square between the CBHs detected by the two devices was 0.814. In autumn, the corresponding adj. R-square was the lowest (0.715) among all four seasons. The adj. R-square values between the CTHs observed by the radiosonde and the MMCR were lower than those of the CBH, indicating that the CTHs detected by the two devices significantly differed. The adj. R-square values between the CTHs detected by the radiosonde and the MMCR in spring and summer were similar: 0.879 and 0.852, respectively. The corresponding adj. R-square in autumn and winter were only 0.629 and 0.710, respectively. Based on the adj. R-square values of the CTH and CBH, the difference between the CVSs was the largest when employing the radiosonde and the MMCR in autumn.

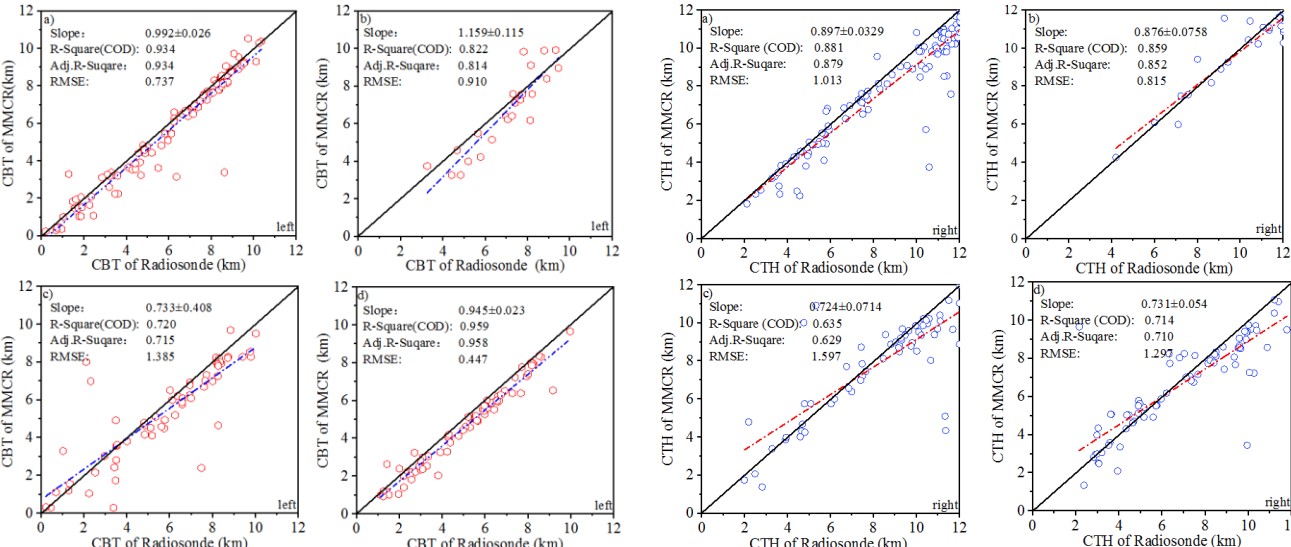

**Figure 12.** Adj. R-square of the CBH ((**left**): red point) and CTH ((**right**): blue point) detection by the radiosonde and MMC R. (**a**) Spring, (**b**) summer, (**c**) autumn, and (**d**) winter. Blue and red dotted lines represent the fitted lines of the CBH and CTH detected by the radiosonde and MMCR, respectively. In the figures, the R-square (COD) is the determination coefficient, and the adj. R-square is the determination coefficient of the modified degree of freedom. The expression of the adj. R-square can be written as $R^2(adj) = 1 - (1 - R^2) \cdot (n-1)/(n-P-1)$, where $P$ is the number of variables, $n$ is the number of samples, and $R$ means R-squared. In univariate linear regression, R-squared and adjusted R-squared assessments are consistent, but the latter is more adaptable to the change in the variables. RMSE refers to the root mean square error.

The distributions of the deviations of the CBHs and CTHs detected by the radiosonde and the MMCR in the four seasons are shown in Figure 13. In spring, the deviation of the CBH (the difference between the CBHs detected by the radiosonde and the MMCR) was mainly distributed between −1 km and 1 km, and the deviation of the CTH was distributed between −1 km and 2 km. In summer, the CBH and CTH deviations were mainly concentrated between 0 km and 2 km and between −0.5 km and 1.5 km, respectively. In autumn, the deviation range and deviation value of the CBHs detected by the radiosonde and the MMCR were large. Both the CBH and CTH deviations were mainly concentrated between −2 km and 2 km. In winter, the CBH and CTH deviations were mainly located between 0 and 1 km and between −2 km and 4 km, respectively.

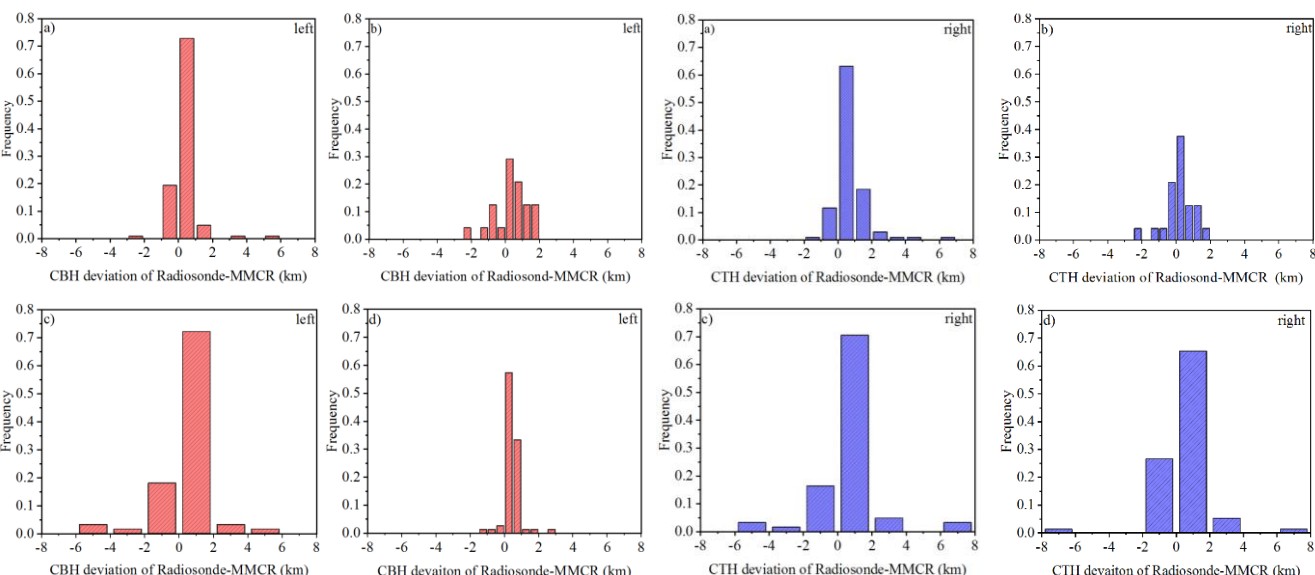

**Figure 13.** Distributions of the CBH (**left**) and CTH deviations (**right**) detected by the radiosonde and MMCR. (**a**) Spring, (**b**) summer, (**c**) autumn, and (**d**) winter.

### 4.2.3. Annual Distributions of Cloud Base and Top Heights

The adj. R-square values of the CBHs and CTHs detected by the MMCR and the radiosonde from December 2020 to November 2021 are shown in Figure 14. The adj. R2 between the CBH detection results of the two devices is 0.880 and the root mean square error (RMSE) is 0.908. The adj. R-Square and RMSE of the CTH results are 0.788 and 1.268, respectively. Figure 15 shows that the CBH deviations detected by the two devices are distributed between −6 km and 6 km, and mainly concentrated between −1 km and 2 km. The CTH deviation distribution is between −8 km and 8 km, and the deviation is mainly concentrated between −2 km and 3 km. The adj. R-square, RMSE, and deviation distribution characteristics of the CBHs detected by the radiosonde and the MMCR show that employing these two devices to detect the CBH achieved a better consistency than in the detection of the CTH.

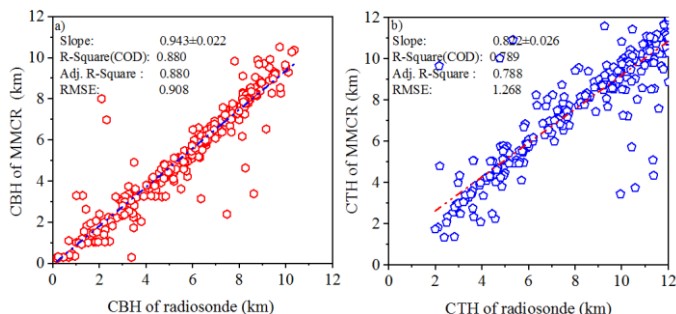

**Figure 14.** Adj. R-square of CBHs and CTHs detected by the radiosonde and MMCR from December 2020 to November 2021. (**a**) The blue dotted line shows the fitted line of CBHs detected by the radiosonde and MMCR, and (**b**) the red dotted line is the fitted line of CTHs detected by the radiosonde and MMCR. In the figures, the R-square (COD) is the determination coefficient, the adj. R-square is the determination coefficient of the modified degree of freedom, and the RMSE is the root mean square error. The expression of the adj. R-square can be written as $R^2(adj) = 1 - (1 - R^2) \cdot (n - 1)/(n - P - 1)$, where $P$ is the number of variables, $n$ is the number of samples, and $R$ means R-squared.

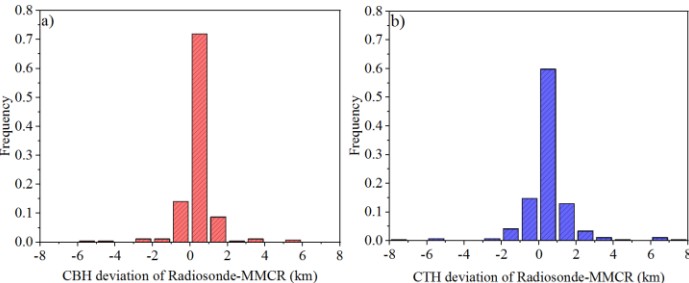

**Figure 15.** Distributions of the deviations of the CBHs and CTHs detected by the radiosonde and MMCR from December 2020 to November 2021. (**a**) CBH deviation and (**b**) CTH deviation.

## 5. Deviation Analysis of Cloud Top Height

It can be seen from the annual distribution of the CTH deviations detected by the radiosonde and the MMCR that, within the range of 0–4 km, the frequency of the deviation distribution is 0.78. Therefore, in most cases, the CTH determined from the RH observed by the radiosonde using the RH threshold was higher than that derived by the MMCR, and its main influencing factors are as follows:

1.　Attenuation and limited sensitivity of MMCR

With the increasing detection distance, the attenuation correction of the MMCR increases. However, for a cloud signal with a weak reflectivity factor, particularly in the cloud area below $-20$ dBZ, there is no great difference between the reflectivity factors before and after the correction. Concurrently, the CTH after the correction is similar to that before the correction, i.e., the CVS remains almost unchanged. In this study, based on the detection performance of the MMCR, the threshold value of the reflectivity factor used to identify a cloud area was set as $-40$ dBZ (a reflectivity factor $> -40$ dBZ indicates a cloud). This shows that the MMCR sensitivity of the cloud detection was sufficient. Although precipitation clouds considerably attenuate an MMCR, the 467 samples used in this study did not include precipitation clouds. Therefore, it can be inferred that the attenuation of the MMCR is not the main cause of the underestimation of the CTH. The limitation of the MMCR detection sensitivity, particularly for high-level cirrus cloud tops, where small ice crystal particles are distributed, is the main cause of its CTH underestimation.

2.　Radiosonde humidity sensor delay

The humidity sensor of the GTS11 radiosonde is a carbon film humidity-sensitive resistor. Its response to a temperature change is gradual at low temperatures (particularly below $-50\,°$C), and the temperature measurement is rather poor. The average CTH detected by the MMCR was approximately 8 km, whereas that detected by the radiosonde based on the RH, using the RH threshold method, was higher than 8 km. When the cloud top temperature was below $-50\,°$C, the RH decreased very gradually with the increase in the detection height. Therefore, the response delay of the humidity sensor caused the CTH overestimation by the radiosonde compared to the results obtained the by MMCR.

3.　Drift in the rising trajectory of the radiosonde

According to the existing literature [7,8,13–15,18,19], the drift of a radiosonde is the main source of the deviation affecting its CTH and CBH measurement. However, these studies do not explain and analyse the deviation caused by the drift in detail. The rising trajectory of a radiosonde takes time to develop, during which the wind speed, wind direction, and turbulence cause it to seriously deviate from the vertical route. Consequently, when the spatial structure of a cloud changes rapidly, differences in the CVS observations obtained using a radiosonde and an MMCR are expected.

Figure 16 shows the distributions of the horizontal wind speed and wind direction experienced by the radiosonde on its rising trajectory over the four seasons (from December 2020 to November 2021: winter, spring, summer, and autumn). In spring and autumn, the wind speed and direction presented similar distribution characteristics. The wind direction

was mainly concentrated in the NNW–SSW directions, and the wind speed below 9 km was less than 70 m·s$^{-1}$, which indicates that the horizontal drifts were similar in spring and autumn. The wind direction scale was large, mainly in the ENW–SW direction, whereas the wind speed scale was small, and the wind speed below 9 km was less than 50 m·s$^{-1}$ in summer. In winter, the wind direction was concentrated mainly in the WNW–SW direction, the wind speed almost reached 90 m·s$^{-1}$ at 9 km, and the radiosonde deviated further from the vertical path under a strong westerly wind.

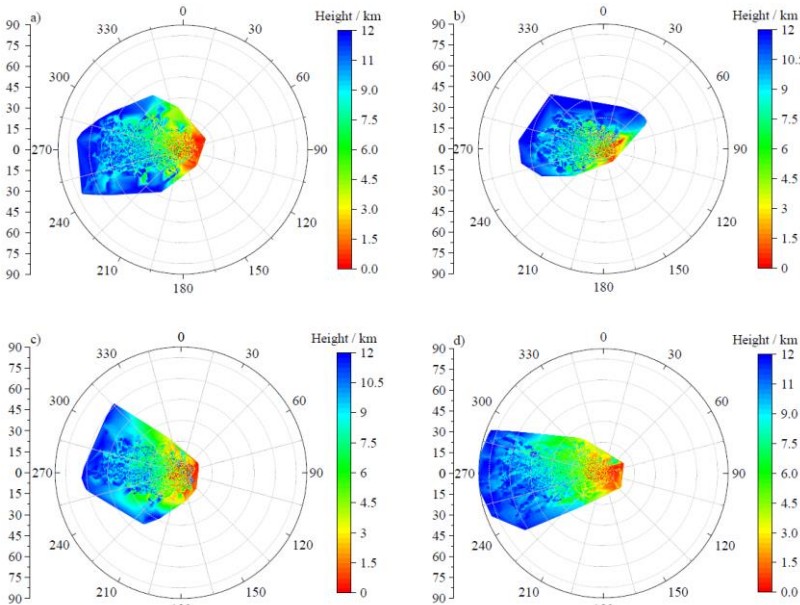

**Figure 16.** Seasonal variation characteristics of the wind speed and direction in the rising trajectory of the radiosonde. (**a**) Spring, (**b**) summer, (**c**) autumn, and (**d**) winter (0°: north, N; 90°: east, E; 180°: south, S; 270°: west, W).

The wind direction and speed are the main factors affecting the rising trajectory of a radiosonde. Figure 17 shows the distributions of the horizontal drifts calculated using the longitude and latitude deviations of the radiosonde relative to the launch point over the four seasons. As described above, the wind direction distribution ranges and wind speeds in spring and autumn were similar; therefore, the horizontal drift distributions were also very similar. Among the four seasons, in summer, the horizontal drift was the lowest, and the average drift at 12 km was only 38 km. In winter, the horizontal drift was the highest, and the relative drift at a height of 9 km reached 38 km. If a cloud structure is loose during the radiosonde observation period, it is probable that it will be different from the cloud observed by the MMCR, owing to the large horizontal drift of the former. In the case of a large-scale and compact cloud, even if there is a large horizontal drift, the radiosonde and the MMCR may detect the same cloud. Therefore, when comparing and analysing the CVSs observed by the two devices, it is necessary to consider the cloud type, cloud height, and horizontal drift in order to provide a more comprehensive and scientific explanation for the differences between their results.

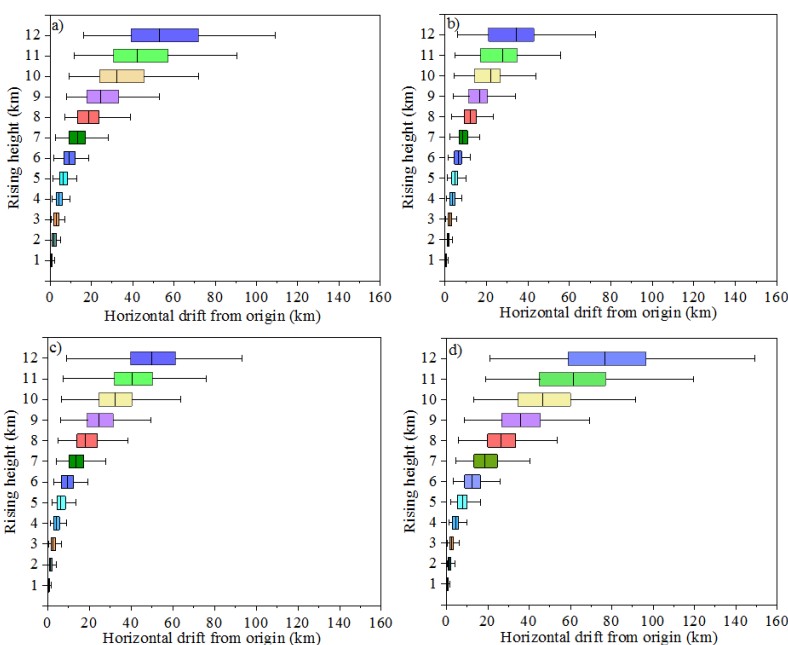

**Figure 17.** Horizontal drift of the radiosonde in its rising trajectory. (**a**) Spring, (**b**) summer, (**c**) autumn, and (**d**) winter (each boxplot shows the minimum, first quartile, median, third quartile, and maximum).

According to the cloud data from Xi'an, cirrus clouds occur mainly in the spring. Cirrus clouds are thin, dense, and mainly composed of non-spherical ice crystal particles, and their horizontal range can reach hundreds to thousands of kilometres. In summer, uniform episodic clouds are dominant, with a horizontal scale of hundreds of kilometres. In autumn, isolated and scattered altocumulus clouds dominate. Greyish-white high-level and stratocumulus clouds are common, and the scale of the clouds is large in winter. The average height of a cloud layer in spring varies between 5.8 and 8.1 km. Within this height range, the horizontal drift reaches approximately 10–20 km. For large-scale cirrus clouds, the horizontal drift is insufficient to cause the radiosonde to pass into other clouds (horizontal drift is not the main factor causing the differences in the CVSs detected by the radiosonde and the MMCR). Moreover, the CTHs detected by the two devices are highly consistent (the adj. R-square of the CBH and CTH are 0.934 and 0.879, respectively, in Figure 13. In summer, the clouds are high, mainly distributed between 6.8 km and 9.8 km, and the drift of the radiosonde is approximately 13–20 km within this range. Compared to the horizontal scale of episodic clouds, this drift is insufficient to cause the radiosonde to float into other clouds (the adj. R-square of the CBH and CTH are 0.814 and 0.852, respectively). In autumn, the average height of the clouds is 5.4–8.0 km, and the drift is 10–18 km. For small-scale clouds that are isolated and dispersed, the radiosonde drifts easily into other clouds during the rising process; i.e., in most cases, both the radiosonde and MMCR pass through the base of the same cloud (the adj. R-square of the CBH is 0.715). However, the radiosonde penetrates the other cloud top, resulting in a low correlation between the CTHs detected by the two devices (the adj. R2 of the CTH is 0.629). In winter, clouds are mainly distributed between 4.8 km and 6.8 km, and the corresponding drift is approximately 10–18 km. In general, for large-scale, high-level and stratocumulus clouds, both the radiosonde and MMCR detect the same clouds. Therefore, they show high consistency in the CBHs (the adj. R-square is 0.958) and CTHs (the adj. R-square is 0.710).

## 6. Statistics and Analysis of the CVS Characteristics in the Xi'an Area

The CVSs observed by the radiosonde and the MMCR are in good agreement, using the adjusted RH threshold and spatiotemporal matching criteria, respectively. In this study, this RH threshold was used to identify cloud information from the RH profiles

recorded by the radiosonde in Xi'an from 2019 to 2021. The objective was to examine the distribution and change characteristics of the CVSs and provide effective supporting data for climate research in Xi'an. Based on Table 3, the highest cloud-free condition in winter was 34.36%, and rainfall occurred most frequently (12.99%) in autumn. In summer, two- and three-layer clouds were the most frequent (22.10% and 5.23%, respectively) in all four seasons, indicating that warm air was conducive to the formation of clouds. In autumn, the frequency (0.75%) of the four-layer clouds was the largest, which may be due to the vast horizontal drift of the radiosonde and the small scale of the clouds, resulting in the highest frequency of the multi-layer clouds identified by radiosonde. The occurrence frequencies of one-, two-, and three-layer clouds in spring and winter were similar.

Because the humidity sensor of the radiosonde responds sluggishly to the RH change within a height range below −50 °C, it overestimates the CTH (see the Appendix A for details). Based on the results shown in Figures A1 and A2, when the humidity sensor was delayed at a temperature below −50 °C, the altitude corresponding to −50 °C was taken as the CTH height detected by the radiosonde. The RH threshold method was used to determine the cloud information from the RH data acquired by the radiosonde from 2019 to 2021. The monthly trends in the average CBH and CTH are shown in Figure 18. The blue line in Figure 18a represents the average CTH variation with the months. The average CTH fluctuated in the range of 7–10 km over the three years. In Figure 18b, the red line shows the change in the average CBH with the months, and it fluctuated within the range of 3–5 km. However, although the CBH and CTH fluctuated slightly with the changing of the months, their changes did not vary significantly over the years.

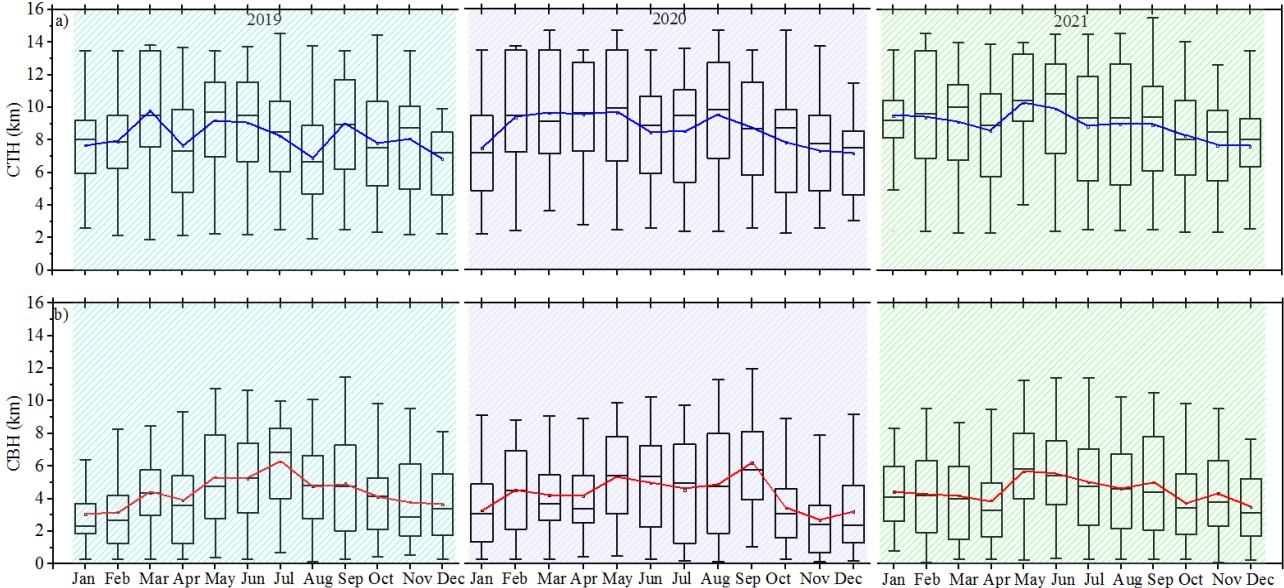

**Figure 18.** Variation trends in the average CTH and CBH for each month from 2019 to 2021. (**a**) Blue represents the average CTH and (**b**) red represents the average CBH (each boxplot shows the minimum, first quartile, median, third quartile, and maximum).

**Table 3.** Cloud distribution from 2019 to 2021.

| Season | Month | Sample Size | Cloud-Free Sample Size/Frequency | Precipitation Cloud Sample Size/Frequency | One-Layer Cloud Sample Size/Frequency | Two-Layer Cloud Sample Size/Frequency | Three-Layer Cloud Sample Size/Frequency | Four-Layer Cloud Sample Size/Frequency |
|---|---|---|---|---|---|---|---|---|
| Spring | Mar, Apr, May | 510 | 153, 30.00% | 51, 10.0% | 224, 43.90% | 71, 13.90% | 10, 1.96% | 1, 0.19% |
| Summer | Jun, Jul, Aug | 525 | 116, 22.10% | 62, 11.81% | 200, 38.10% | 116, 22.10% | 28, 5.23% | 3, 0.57% |
| Autumn | Sep, Oct, Nov | 531 | 169, 31.83% | 69, 12.99% | 179, 33.71% | 98, 18.46% | 12, 2.26% | 4, 0.75% |
| Winter | Dec, Jan, Feb | 521 | 179, 34.36% | 28, 5.37% | 232, 44.53% | 70, 13.44% | 10, 1.92% | 2, 0.38% |

## 7. Conclusions

In this study, the adjusted RH threshold method was used to identify CVSs from RH profiles recorded by a radiosonde. Based on the established spatiotemporal matching criteria, CVSs detected by an MMCR and the radiosonde were compared and analysed. The deviation of the CVSs detected by the two devices was analysed in detail in terms of the cloud type, cloud height, and horizontal drift of the radiosonde. Finally, using the radiosonde data from 2019 to 2021, the CVSs over Xi'an were statistically analysed. The main conclusions of this study are as follows:

1. The adjusted RH threshold method effectively identified cloud information from the RH profiles recorded by the GTS11 radiosonde in Xi'an. Spatiotemporal matching criteria can effectively reduce the detection deviation of the CVSs caused by the horizontal drift of the radiosonde.

2. The GTS11 radiosonde and MMCR results showed high consistency in the observation of the CVSs of low-level clouds. However, with the increase in the cloud height, the frequency of clouds detected by the radiosonde became higher than that detected by the MMCR.

3. In summer, large-scale clouds were distributed at high heights, and the radiosonde experienced a wide range of wind directions and a low wind speed during the rising process, resulting in a low horizontal drift. Therefore, the CVS results of the radiosonde and the MMCR were similar. The cloud height distributions in spring and autumn were similar, causing the wind speed and direction distributions of the radiosonde on the rising trajectory to be similar. Therefore, the drift was approximately the same, whereas the cloud size in autumn was small, and the correlation between the CVS observations by the radiosonde and the MMCR was lower than that observed in summer. In winter, the concentrated wind direction and high wind speed caused a large drift. However, the cloud height was low and its size was large. Thus, there was no significant difference between the CVSs detected by the two devices. Therefore, when using the RH threshold method to identify a CVS from radiosonde RH profiles, not only the horizontal drift of the radiosonde, but also the cloud type and cloud height, should be considered.

4. In different seasons, the cloud types, cloud height, horizontal drift of radiosonde, and the delay of humidity sensor were the main factors affecting the accuracy of the radiosonde in detecting the CVSs. Although the MMCR was subject to some limitations when detecting precipitating clouds and high-level cirrus clouds, it could remove near-surface moist layers with no clouds. The CVSs distribution and change characteristics examined in this study can provide better support for the numerical model analysis and study of climate change characteristics in Xi'an.

5. Using the RH threshold method to identify CVSs from radiosonde RH profiles from 2019 to 2021 in Xi'an showed that the cloud-free condition was the highest (34.36%) in winter, and precipitation clouds appeared most frequently (12.99%) in autumn. The frequencies of two-layer (22.10%) and three-layer (5.23%) clouds were the highest in summer. The average CTH and CBH did not fluctuate significantly with the changing of the years. The average CTH and CBH fluctuated in the ranges of 7–10 km and of 3–5 km, respectively.

**Author Contributions:** Conceptualization, Y.Y.; investigation, Y.Y.; methodology, Y.Y. and H.D.; software, Y.Y.; supervision, H.D. and D.H.; data collation and summary, Y.L., D.C., N.C. and Q.Y.; writing—original draft, Y.Y.; writing—review & editing, Y.Y. and H.D.; project administration, D.H. All authors have read and agreed to the published version of the manuscript.

**Funding:** This research was supported by the National Natural Science Foundation of China, Innovative Research Group Project of the National Natural Science Foundation of China (grant Nos. 42130612 and 41627807), and the Ph.D. innovation fund projects of the Xi'an University of Technology (grant No. 310-252072106).

**Data Availability Statement:** The data and code related to this article are available upon request from the corresponding author.

**Acknowledgments:** We express our thanks to the Xi'an Meteorological Bureau, Shehong Li and Shuicheng Bai, for providing the radiosonde data.

**Conflicts of Interest:** The authors declare that they have no conflicts of interest related to this work.

## Appendix A. Analysis of the Radiosonde Delay

The difference between the CTHs detected by the radiosonde and the MMCR is generally considered to be caused by two factors: (1) The MMCR cannot respond to small particle information at a cloud top, owing to its limited detection sensitivity, resulting in the CTH underestimation. (2) The CTH is overestimated by the radiosonde owing to the delay of its humidity sensor at temperatures $< -50\ ^{\circ}$C. Compared to the MMCR, a lidar has a better detection ability for small particles; therefore, we used lidar data in the same period to verify factor (1): whether the underestimation of the CTH is caused by the limited sensitivity of the MMCR. Figure A1a shows the results of the cloud information obtained from the radiosonde RH profile on 7 March 2021, using the RH threshold method. Accordingly, the CTH is 10.3 km. The average CTHs detected by the MMCR and lidar are similar in this period: 8.6 km (Figure A1b) and 8.4 km (Figure A1c), respectively. Therefore, in this case, the CTH detected by the MMCR is more accurate than that detected by the radiosonde. The temperature of the GTS11 radiosonde is $-50\ ^{\circ}$C at 9.2 km, and it decreases gradually with the increase in the cloud height. Therefore, it can be inferred that the information between 9.2 km and 10.3 km is misidentified as a cloud by the radiosonde, owing to the response delay of its humidity sensor, resulting in the overestimation of the CTH.

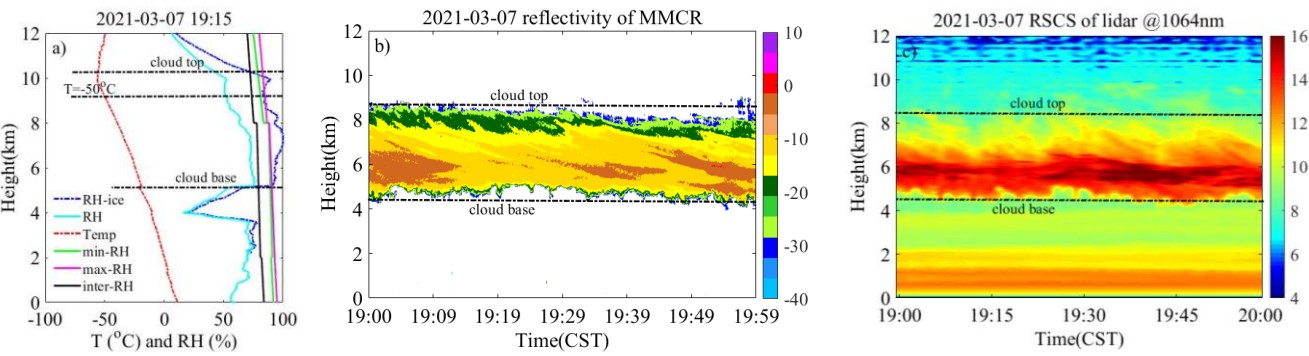

**Figure A1.** Cloud information observed by the radiosonde, MMCR, and lidar on 7 March 2021. (**a**) Temperature and RH obtained by the radiosonde, (**b**) reflectivity factor observed by the MMCR, and (**c**) 1064 nm signal of the lidar.

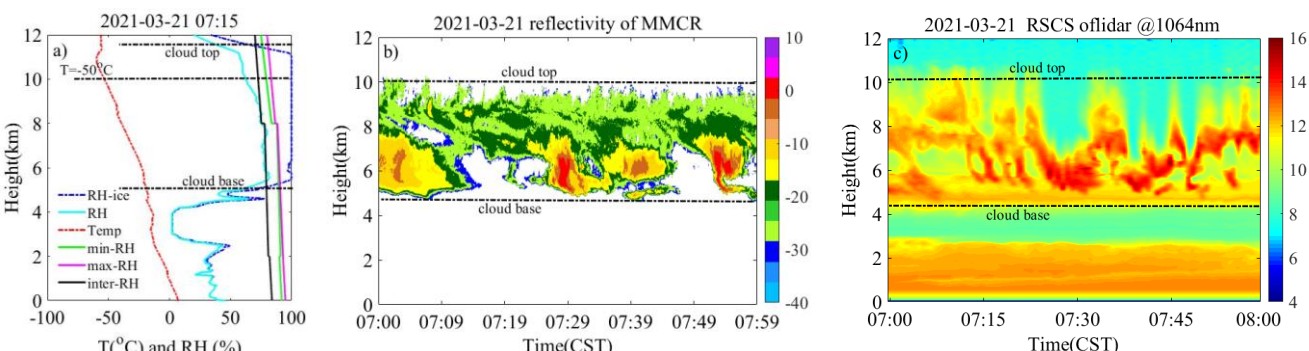

**Figure A2.** Cloud information observed by the radiosonde, MMCR, and lidar on 21 March 2021. (**a**) Temperature and RH obtained by the radiosonde, (**b**) reflectivity factor observed by the MMCR, and (**c**) 1064 nm signal of the lidar.

Figure A2 shows the same plots as Figure A1 but for 21 March 2021. The radiosonde overestimates the CTH, owing to the delayed response of its humidity sensor. The CTH detected by the radiosonde is 11.5 km. The average CTHs detected by the MMCR and the lidar are similar: 9.8 km and 9.9 km, respectively. Therefore, the CTH, in this case, is ~9.9 km. The height corresponding to the radiosonde temperature of $-50\,°C$ is 10.0 km, and the RH within the range of 10.0–11.5 km exceeds the RH threshold, and the wet layer is incorrectly identified as a cloud. Therefore, the cloud top is overestimated by 1.6 km by the radiosonde. Among the scenarios shown in Figures A1 and A2, 19% of cases (data from December 2020 to November 2021) are CTH overestimations due to the delay of the humidity sensor.

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
