# Peer review of "Confidence and Error Analyses of the Radiosonde and Ka-Wavelength Cloud Radar for Detecting the Cloud Vertical Structure"

_remotesensing, doi:10.3390/rs14184462_

Round 1
Reviewer 1 Report
Millimeter wave radar and sounding balloon are important equipment for detecting the vertical profile of the atmosphere and cloud. This manuscript evaluate the reliability and stability of using RH threshold method to identify cloud information by using Ka-band millimeter-wave cloud radar (MMCR) observation data. By comparing and analyzing the observation results, it is concluded that radiosonde and MMCR have high consistency in the detection of low-level structures. the differences between the results of different equipment are also analyzed in detail. The manuscript is rich in content and the research results are also of great significance. It has a certain reference value for people to identify clouds,. The manuscript can be accepted for publication after minor revision.
The paper analyzes the difference of cloud vertical structure for different types of clouds detected by sounding balloon and millimeter wave radar. Does the application of millimeter wave radar data improve the quality of cloud layer identification by sounding balloon data? Especially for the identification of cloud tops, can a standard be proposed for the determination of two different types of data?
Author Response
Response to Reviewer 1 Comments
Dear Reviewer.
We greatly thank you for the thorough and valuable suggestions to our work. The manuscript has been polished and modified by professional organizations, and English has been greatly improved. We have made a point-to-point response to these opinions and suggestions, and believe that the quality of the manuscript has been promoted now. All comments have been modified and added in the revised manuscript (mark with red font), and the responses to each comment are given below.
Thank you very much for considering our work!
Yours sincerely,
All authors
%%%
Comments and Suggestions for Authors
Millimeter wave radar and sounding balloon are important equipment for detecting the vertical profile of the atmosphere and cloud. This manuscript evaluate the reliability and stability of using RH threshold method to identify cloud information by using Ka-band millimeter-wave cloud radar (MMCR) observation data. By comparing and analyzing the observation results, it is concluded that radiosonde and MMCR have high consistency in the detection of low-level structures. The differences between the results of different equipment are also analyzed in detail. The manuscript is rich in content and the research results are also of great significance. It has a certain reference value for people to identify clouds. The manuscript can be accepted for publication after minor revision.
Point 1: The paper analyzes the difference of cloud vertical structure for different types of clouds detected by sounding balloon and millimeter wave radar. Does the application of millimeter wave radar data improve the quality of cloud layer identification by sounding balloon data? Especially for the identification of cloud tops, can a standard be proposed for the determination of two different types of data?
Response 1:
1) The joint detection of radiosonde and MMCR will improve the identification accuracy of the CVS. When the humidity of the underlying moist layer is too high, the radiosonde will misjudge the moist layer as a cloud. In this case, the cloud layer detected by MMCR can be used to improve the measurement accuracy of radiosonde. For high-level cirrus clouds, MMCR cannot effectively detect small particles on the cloud top due to the limited detection sensitivity, which will underestimate the cloud top height. At this time, the cloud top height detected by radiosonde detection ((excluding the height distance misjudged due to the response delay of the humidity sensor) is closer to the real cloud top height.
2) For the humidity sensor mounted on the CTS11 radiosonde, if the cloud top height detected by radiosonde is higher than the height derived from MMCR, and temperature at the cloud top is less than -50 ℃, the height detected by radiosonde can be taken as the effective cloud top height. In this case, MMCR cannot obtain the effective cloud top height information effectively.

Reviewer 2 Report
the paper can be accepted after some minor grammar corrections and English edits and scientific writing. For example, in the abstract:please define what adj. r square is. Why not correlation coefficient?
line 83: its better to write detection of clouds using or by Radiosonde measurements. same in the next section.
The paper is consistent throughout.
Author Response
Dear Reviewer.
We greatly thank you for the thorough and valuable suggestions to our work. The manuscript has been polished and modified by professional organizations, and English has been greatly improved. We have made a point-to-point response to these opinions and suggestions, and believe that the quality of the manuscript has been promoted now. All comments have been modified and added in the revised manuscript (mark with red font), and the responses to each comment are given below.
Thank you very much for considering our work!
Yours sincerely,
All authors
%%%
Comments and Suggestions for Authors
Point 1: The paper can be accepted after some minor grammar corrections and English edits and scientific writing. For example, in the abstract: please define what adj. r square is. Why not correlation coefficient?
Response 1:
1) The manuscript has been edited and embellished by a professional editing organization. It is believed that the quality of the manuscript will be greatly improved.
2) Adj. R-square is defined and explained in the manuscript.
R-squared (value range [0-1]) describes the degree of interpretation of input variables to output variables. In univariate linear regression, the larger the R-squared, and the better the fitting degree. Its expression is:
R2= 1−RSS/TSS, (1)
where TSS is the inherent variance of response variables, RSS is the sum of the squares of the residuals. However, as long as more variables are added, whether or not the added variables are related to the output variables, R-squared will either remain unchanged or increase. Therefore, it need to be described with the adjusted R-squared (range (− ∞, 1]). The expression of Adj. R-square can be written as
R2(adj)=1− (1−R2)*(n−1)/(n− P −1), (2)
where P is the number of variables, n is the number of samples. In univariate linear regression, R-squared and adjusted R-squared assessments are consistent, but the latter is more adaptable to the change of variables.
Point 2: line 83: Its better to write detection of clouds using or by Radiosonde measurements. same in the next section.
Response 2:
Line 83: “Radiosonde detects cloud” is modified ad “Clouds detection by radiosonde”
Line 113: “MMCR detects cloud” is modified ad “Clouds detection by MMCR”
Point 3: The paper is consistent throughout.
Response 3:
We carefully edited and checked the manuscript to make it more coherent and fluent.

Reviewer 3 Report
see Review file

Author Response
Dear Reviewer.
We greatly thank you for the thorough and valuable suggestions to our work. The manuscript has been polished and modified by professional organizations, and English has been greatly improved. We have made a point-to-point response to these opinions and suggestions, and believe that the quality of the manuscript has been promoted now. All comments have been modified and added in the revised manuscript (mark with red font), and the responses to each comment are given below.
Thank you very much for considering our work!
Yours sincerely,
All authors
%%%
Review of Remote Sensing 1874612
This paper does a thorough comparison between measurements made at Xi’an city in China by a Ka radar and RH probe on a radiosonde. The purpose is to show that RH threshold cloud identification method appears to give useful results that can be related to global climate change. The paper is not acceptable in its present form. The main issue is the use of the English language which needs review by an individual fluent in writing English.
We greatly thank you the reviewer for the thorough and valuable suggestions to our work. The manuscript has been polished and modified by professional organizations, and English has been greatly improved.
Comments:
Point 1: The authors need to explain how their measurement approach and results can be related to climate change in one city given the demonstrated significant variability in their measurements, as well as the described uncertainties in both measurements. What amount of climate change would correspond to this variability/uncertainty, and is this hypothetical change reasonable? If this cannot be answered, it is recommended that the comments dealing with climate.
Response 1: Both lines 30-34 and 66-69 in the manuscript mention that cloud changes affect climate change. We want to express that cloud play an important role in atmospheric radiation transmission and affect climate change. The CVS is the main factor that affects the change of radiation transmission. Therefore, we hope to provide effective data support for climate change in Xi'an by analyzing the CVS change and distribution characteristics. At present, we can't explain in detail how the measurement results will affect the climate change in Xi’an city, which is also supplemented and explained in the conclusion of the manuscript.
Point 2: The paper nicely illustrates the complex nature of clouds as measured by vertical location, horizontal extent, type and thickness. Given this complexity, the paper also illustrates the difficulty of applying the “RH threshold method” for identifying clouds with the RH probe on the radiosonde. The latter should be considered as a main conclusion of the paper.
Response 2: Corresponding conclusions have been added. See response 4 for details.
Point 3: The content, such as strengths and weaknesses, dealing with Wang and Rossow, Zhang et al, and the paper’s variation of the latter RH threshold method should be more clearly organized and described.
Response 3: Lines 52-64 in the introduction are re-described as:
Wang and Rossow [12] found that the RH change of radiosonde was closely related to cloud information. Through frequency statistics of the RH value within the cloud base height range observed by the ground and radiosonde for one year, they proposed to take 84 % - 87 % as the threshold value, and judge the CVS under the condition of the negative positive jump of the RH generated by the cloud top and cloud base, that is, the RH threshold method (WR95), However, WR95 tended to misclassify moist cloudless atmospheric layers as clouds, and both radiosonde techniques reported higher cloud tops than those from cloud radar. Zhang et al. [19] used the radiosonde data (Vaisala radio- sonde-RS92) observed by the Atmospheric Radiation Measurement Mobile Facility was deployed in Shouxian, Anhui Province, China, (116°27′-117°04′ E, 31°54′-32°40′ N, subtropical northern monsoon humid climate), and combined with the observation results of MMCR and cloud altimeter to improve WR95. Although the influence of horizontal drift of radiosonde is considered, the reliability of radiosonde detection under different weather conditions and different cloud types is not analyzed. Therefore, when analyzing the accuracy and reliability of radiosonde detecting the CVS, it is necessary to comprehensively consider the horizontal drift caused by wind speed and wind direction of radiosonde and the changes of different weather and different cloud types, so as to comprehensively evaluate the performance of radiosonde detecting the CVS.
Point 4: The Conclusions Section is insufficient since it only repeats some of the measurement. Itemize the results from your study. E.g.: climate change can be (cannot be) determined by this approach; rain limits the accuracy of the radar, response time and radiosonde drift limit the RH measurement, etc.
Response 4: Add the corresponding conclusion as follows:
4) In different seasons, cloud types, cloud height, horizontal drift of radiosonde and delay of humidity sensor are the main factors affecting the accuracy of radiosonde in detecting the CVS. Although MMCR has some limitations in detecting precipitating clouds and high-level cirrus clouds, it can remove near-surface moist layers with no clouds. The CVS distribution and change characteristics studied in this study, which can provide better support for numerical model analysis and study of climate change characteristics in Xi'an.
Point 5: In Title – You say “... for cloud vertical structure”. While you identify the cloud types, little is said about cloud structure. Suggestion: say instead ‘…for cloud height and type’
Response 5: The research content includes the cloud base height, cloud top height, cloud layer number, cloud types, and they are collectively referred to as cloud vertical structure. Unfortunately, we have not well-defined the cloud vertical structure in this study, which leads to confusion. Therefore, the cloud vertical structure is redefined in the introduction, which echoes the title of the manuscript.
Point 6: Section 2.3 – Why were the particular times chosen for your “spatiotemporal matching?
Response 6: First of all, the time when MMCR’s beam reaches the clouds is much shorter than the time spent by radiosonde, so there is a certain difference in time between the two detection methods. Secondly, the cloud structure changes in space, such as growth, dissipation and flutter. If the cloud base height and cloud top height detected by MMCR are directly compared with the heights derived from Radiosonde, which will cause great error. Considering the rising speed (time dimension) of Radiosonde and assuming cloud moving direction (space dimension), the spatiotemporal matching is proposed, that is, the average cloud base height and cloud top height in
certain time limit is used as the results of cloud boundary detected by radiosonde.
Point 7: Adj. R-square – This word is used numerous times in the paper with the description that it is a “coefficient adjusted for modified degree of freedom”. Explain what this means, and where this coefficient/plot appears in the paper.
Response 7: Adj. R-square is defined and explained in the manuscript.
R-squared (value range [0-1]) describes the degree of interpretation of input variables to output variables. In univariate linear regression, the larger the R-squared, and the better the fitting degree. Its expression is:
R2= 1−RSS/TSS, (1)
where TSS is the inherent variance of response variables, RSS is the sum of the squares of the residuals. However, as long as more variables are added, whether or not the added variables are related to the output variables, R-squared will either remain unchanged or increase. Therefore, it need to be described with the Adjusted R-squared (range (− ∞, 1]). The expression of Adj. R-square can be written as
R2(adj)=1− (1−R2)*(n−1)/(n− P −1), (2)
where P is the number of variables, n is the number of samples. In univariate linear regression, R-squared and adjusted R-squared assessments are consistent, but the latter is more adaptable to the change of variables.
Point 8: L 86-87 – Define what you mean by “two cloud bases”
Response 8: Line 86-87 expression is not clear, and the re description is as follows:
The WR95 method is described to use rawinsonde data to estimate cloud vertical structure, including cloud-top and cloud base heights, cloud-layer thickness, and the characteristics of multilayered clouds. Cloud –layer base and minimum relative humidity of at least 84%, and relative humidity jumps exceeding 3% at cloud-layer top and base ,where relative humidity is with respect to liquid water at temperatures greater than or equal to 0℃ and with respect to ice at temperatures less than 0℃.
Point 9: L 92 – Describe “merging two adjacent humidity layers” and where and when this happens. This affect appears to have little effect in the plots, so is it necessary to be included?
Response 9: The ‘merging two adjacent humidity layers’ means ‘two contiguous layers are considered as a one-layer cloud if the distance between these two layers is less than 300 m or the minimum RH within this distance is more than the maximum inter-RH value within this distance’.
As shown in the black box in the figure below, the spacing distance between the two moist layers is 7203-6594 = 609m, and this distance has been greater than 300m. However, minimum RH within this distance is more than the maximum inter-RH value within this distance. So, two contiguous moist layers are considered as a one-layer cloud, that is, the cloud base and cloud top heights of the high-level cloud are 6282m and 8226m respectively.
-100-80-60-40-20020406080100T(oC) and RH (%)024681012Height(km)20201212.19:15RH-iceRHTempmin-RHmax-RHinter-RH
Point 10: L 98 – Define the difference between “measurement span” and “effective measurement span”. What is delta(U)?
Response 10: Writing error. ‘Δ(U) ≤ 5% RH’ should modified as ‘Δ(RH) ≤ 5% RH’.
Point 11: L99 – Explain the cause of the variability of RH thresholds with height and temperature
Response 11: Theoretically, the RH is proportional to the water pressure and inversely proportional to the saturated water pressure. With the increase of altitude, the water vapor content in the air decreases, and the RH decreases with the decrease of water vapor content, but the decrease of temperature means that the saturated water vapor pressure decreases. Therefore, under the standard atmospheric conditions, if the water vapor content is unchanged, the RH increases with the elevation rise and the temperature decrease. Similarly, under the condition that the temperature remains unchanged, the RH increases with the increase of water vapor content. In general, under normal atmospheric conditions (or standard atmospheric conditions), the RH decreases with the altitude increase, the temperature decrease and water vapor content decrease.
Point 12: L371 – What is CBT in the red plots?
Response 12: Writing error. ‘Adj. R-Square of CBH (left: red point) and CTH (right: blue red point) detection by radiosonde and MMCR’ should modified as ‘Adj. R-Square of CBH (left: red points) and CTH (right: blue points) detection by radiosonde and MMCR’.
Point 13: L390 – CTH deviation plots are mis-labelled.
Response 13: The figures arrangement are wrong, and the arrangement after correction are as follows:
Fig. 13 Distribution of the CBH (left) and CTH deviation (right) detecting by radiosonde and MMCR. a) spring, b) summer, c) autumn and d) winter.
Point 14: L543 – There is no 4) conclusion
Response 14: Writing error. It has been corrected in the study.
